# The Double Engines and Single Checkpoint Theory of Endometriosis

**DOI:** 10.3390/biomedicines10061403

**Published:** 2022-06-14

**Authors:** Che-Fang Hsu, Aye Aye Khine, Hsuan-Shun Huang, Tang-Yuan Chu

**Affiliations:** 1Center for Prevention and Therapy of Gynecological Cancers, Department of Medical Research, Hualien Tzu Chi Hospital, Buddhist Tzu Chi Medical Foundation, Hualien 970, Taiwan; cfhsu@tzuchi.com.tw (C.-F.H.); aye2224@tzuchi.com.tw (A.A.K.); san.sam@msa.hinet.net (H.-S.H.); 2Department of Obstetrics & Gynecology, Hualien Tzu Chi Hospital, Buddhist Tzu Chi Medical Foundation, Hualien 970, Taiwan; 3Department of Life Sciences, Tzu Chi University, Hualien 970, Taiwan; 4Department of Molecular Biology and Human Genetics Tzu Chi University, Hualien 970, Taiwan; 5Buddhist Tzu Chi General Hospital, 707, Section 3, Chung-Yang Road, Hualien 970, Taiwan

**Keywords:** endometriosis, ovulation, retrograde menstruation, hypermutation, immune checkpoint

## Abstract

Endometriosis is a chronic disease characterized by the ectopic localization of the endometrial tissue in the peritoneal cavity. Consequently, it causes local pathological changes and systemic symptoms, affecting at least one in every ten women. This disease is difficult to diagnose early, it is prone to dissemination, is difficult to eradicate, tends to recur, and is regarded as “a cancer of no kill”. Indeed, the development of endometriosis closely resembles that of cancer in the way of mutagenesis, pelvic spreading, and immunological adaptation. While retrograde menstruation has been regarded as the primary cause of endometriosis, the role of ovulation and menstrual stimuli in the development of endometriosis has long been overlooked. The development of ovarian and peritoneal endometrioses, similar to the development of high-grade serous carcinoma in the fallopian tube fimbriae with intraperitoneal metastasis, depends highly on the carcinogens released during ovulation. Moreover, endometriosis carries an extremely hypermutated genome, which is non-inferior to the ultra-mutated endometrial cancer. The hypermutation would lead to an overproduction of new proteins or neoantigens. Because of this, the developing endometriosis may have to turn on the PD-1/PDL-1 “self-tolerance” checkpoint to evade immune surveillance, leaving an Achilles tendon for an immune checkpoint blockade. In this review, we present the double engines and single checkpoint theory of the genesis of endometriosis, provide the current pieces of evidence supporting the hypothesis, and discuss the new directions of prevention and treatment.

## 1. Endometriosis Behaves Like a Cancer of No-Kill

Endometriosis is a disease wherein an endometrium-like tissue grows ectopically outside the uterus. It affects about 10% of reproductively-active females and causes debilitating dysmenorrhea and pelvic pain, as well as infertility and a substantial social and healthcare burden [1]. The pathobiology underlying this disease is complicated since it is associated with heterogeneous symptoms and affects multiple physiological systems [2,3]. While imaging studies, such as ultrasonography and MRI, can only suggest a possible diagnosis [4,5], surgical visualization and histological verification can establish a reliable diagnosis. Nevertheless, the pathophysiological mechanisms that lead to the development and maintenance of endometriosis remain unclear. Currently, there are available medical therapies which focus on treating the symptoms rather than the causes; thus, efficacy is limited because of the severe side effects that prohibit long-term treatment [6]. Moreover, endometriosis often recurs after discontinuing treatment. Notably, as endometriosis has cancer-like characteristics, such as hypermutation, intraperitoneal spreading, and invasion, and the propensity of its recurrence post-treatment [7], it can be regarded as a cancer of no-kill.

## 2. Enigmas and Missing Links in Endometriosis

Although there is no clear evidence on the etiology of endometriosis, there are multiple hypotheses regarding the origin of endometriotic lesions. The most acknowledged hypothesis is that of retrograde menstruation, wherein there is a backflow of menstrual blood into the peritoneal cavity through the fallopian tubes [8]. Tissue fragments in this retrograde blood-flow seed the ectopic sites on the peritoneal and ovarian surfaces. However, retrograde menstruation is almost universal among menstruating women [9], whereas endometriosis is uncommon. Thus, factors, such as female sex hormones and inflammatory and immunological milieus, may determine the viability and persistence of the endometrial tissue in the ectopic location [6,9,10,11]. Given these risk factors, recent studies have implied missing links between retrograde menstruation and the genesis and progression of endometriosis. In the following sections, we will present three missing links in the pathogenesis of endometriosis and discuss the underlying mechanisms and possible clinical applications in the prevention of disease and targeted therapy.

## 3. Missing Link 1: Ovulation Is an Overlooked Etiological Factor in the Development of Endometriosis

### 3.1. Epidemiological Evidence

Epidemiological studies have revealed multiple risk factors for endometriosis [3,10], which can be grouped into two categories, both associate with ovulation.

(1)The first category is the factors related to incessant ovulation, such as a short length of the menstrual cycle, nulliparity, a short duration of lactation, and no consumption of oral contraceptives [11,12,13,14]. These risk factors are associated with an increased frequency of exposure to the menstrual/ovulation cycle. Multiple lines of evidence indicate that a frequent exposure to both menstrual and ovulation cycles are responsible for the development of endometriosis (detailed in the following section).(2)The second category includes the risk factors associated with exposure to insulin-like growth factor (IGF) axis proteins that are the main effectors of growth hormones (GHs) in pubertal body-height spurt [15]. A low body-mass index and waist-to-hip ratio are two GH/IGF-related risk factors for endometriosis [16,17,18] and are also the two major phenotypes of the GH/IGF axis. The risk of developing endometriosis is reported to be positively associated with a woman’s height and negatively associated with a woman’s weight [19]. Additionally, large-scale genome-wide association studies have reported the positive association of the IGF axis genes, including the *insulin-like growth factor 2 binding protein (IGFBP)2*, *IGF2BP3*, and the IGFBP-cleavage enzyme *PAPP-A*, and *IGF-1R,* with an adult body height [20]. The IGF axis signal is highly abundant and active in the ovulatory follicular fluid (FF), which is physiologically important for the regeneration of the damaged tissue post ovulation [21]. Reportedly, this ovulatory growth signal also plays an essential role in the genesis of ovarian high-grade serous carcinoma (HGSC), conferring for the stemness activation, clonal expansion and anchorage-independent growth phenotypes (detailed in the later section) [21].

### 3.2. Clinical Evidence

Incessant ovulation cycles, as a risk factor for the development of endometriosis, can be explained by (1) a repeated exposure to ovulatory follicle contents, and (2) an exposure to sex hormones, such as estrogen, or (3) retrograde menstruation. While these factors are always linked together, there are clinical scenarios where they are dissociated to indicate their individual role in the development of endometriosis:(1)A use of combined oral contraceptives, which inhibits ovulation while maintaining the menstruation cycles, reduces the risk of both the new diagnosis [22] and a recurrence of endometriosis [23]. Thus, ovulation per se is associated with the development of endometriosis.(2)There are studies on a contrary scenario, where only menstruation is inhibited, further indicating that menstruation is not an essential factor in the recurrence of endometriosis. Women who have the levonorgestrel-releasing intrauterine system (Mirena) typically experience amenorrhea due to atrophy of the endometrium after continuous exposure to a high level of levonorgestrel. However, the systemic concentration (218 ng/L) is insufficient to inhibit ovulation [24]. In a randomized trial where Mirena vs. placebo was tested for the maintenance therapy after endometriosis surgery, Mirena was found to be ineffective in preventing the recurrence [25]. Thus, mitigating menstruation while keeping the cyclic ovulation fails to curb the development of endometriosis, further highlighting the role of ovulation.(3)Upon removal of the uterus (the source of retrograde menses), it is the presence or absence of the ovary that significantly affects the risk of recurrence of endometriosis after surgical treatment [26]. A meta-analysis comparing a hysterectomy-based treatment of endometriosis/adenomyosis to an oophorectomy-based treatment reported an eight-fold increase in the risk of reoperation and a six-fold increase in the risk of recurrent pain when the ovary was preserved [27]. The oophorectomy-based treatment abolished not only ovulation but also estrogen production, which either may be responsible for the prevention of recurrence. However, the following clinical scenario suggested estrogen plays a non-sufficient role.(4)Although the gonadotropin-releasing hormone agonist (GnRHa) suppresses all the supposed risk factors of endometriosis, i.e., ovulation, menstruation, and estrogen, it is only as effective as oral contraceptives in treating endometriosis [28]. Moreover, due to the severe side effects of hormone deprivation, the addition of estrogen to GnRHa therapy is commonly practiced. If estrogen is sufficient for the genesis of endometriosis, this practice would lead to the recurrence of the disease. However, studies have reported that estrogen add-back does not compromise the efficacy of disease control [29,30,31]. Thus, ovulation and menstruation factors seem to be more important than estrogen alone.(5)Lastly, the exceptionally high efficacy of treating endometriosis by dienogest or progesterone-only oral contraceptive pills, which suppresses ovulation and eventually induces amenorrhea, further enforces the theory that retrograde menstruation and ovulation are the major factors that govern the development of endometriosis.

### 3.3. Ovulatory Follicular Fluid Draining to the Peritoneal Cavity May Power the Development of Endometriosis

Previously, in vitro, in vivo, and clinical research on the etiology of ovarian high-grade serous carcinoma (HGSC) has implicated that incessant ovulation [32] is responsible for the development of ovarian HGSC originating from the fallopian tube fimbrial epithelium (FTE) [33]. Similar to the endometriotic lesions, the fimbriae, which pick up oocytes, are exposed to the ovulatory follicular fluid (FF) during and after ovulation. The FF carries growth factors, such as IGF2 and hepatocyte growth factor (HGF), to ensure the regeneration of damaged fimbrial and ovarian epithelia after ovulation, as well as the malignant transformation [21,34,35,36]. Ovulation sources the follicular ingredients into the peritoneal fluid and constitutes the main contents of the peritoneal fluid, which is regarded as an ovarian exudate [37,38]. The epithelium of the endometriosis lesion expresses high levels of IGF-1R and c-MET, the receptor for IGF1/2 and HGF, respectively [39,40]. Previous studies have found a higher level of IGF [41] and HGF [39] in the peritoneal fluid from endometriosis patients than in those without endometriosis, and the level of HGF correlated positively with the revised American Society of Reproductive Medicine score. In addition to the growth factors, FF also sources the key endometriosis-related sex hormone, estrogen, into the peritoneal fluid [42]. Overall, ovulation can be an engine that powers the development of endometriosis in the peritoneal cavity.

### 3.4. Ovulatory Follicular Fluid Induces Cell Stemness, Clonal Expansion, and Malignant Transformation

The mechanism by which the FF promotes tissue regeneration and malignant transformation has been elucidated recently [21,43]. As summarized in Figure 1, two growth signaling pathways are involved. The IGF signaling pathway is responsible for stemness activation and clonal expansion, whereas the HGF signaling pathway is responsible for cell proliferation, migration, and invasion. In brief, the FF harbors abundant IGF2 molecules that are trapped by the IGF binding proteins (IGFBP2 and IGFBP6). Upon ovulation with a follicle rupture, the lytic enzyme PAPP-A is activated to cleave the IGFBPs and release IGF2 to bind IGF-1R on the membrane of fallopian tube fimbria epithelial cells. The subsequent signaling of the AKT/NANOG and AKT/mTOR pathways activates cell stemness and induces clonal expansion, respectively [21]. By being exposed to the same FF after ovulation, the endometriosis cells that express IGF-1R may also respond with a growth signal with the clonal expansion of stem/progenitor cells [43].

### 3.5. Activation of the Coagulation/HGF Cascade in the FF Sustains Cell Transformation Activities after Ovulation

The HGF is the second oncogenic growth factor in FF. Upon binding to c-MET on the FTE, HGF promotes activities related to malignant transformation, including cell proliferation, migration, invasion, and anchorage-independent growth. Unlike the IGF axis, which functions transiently after ovulation, the activity of the HGF in ovulatory FF is sustained long after its drainage into the peritoneal fluid throughout the menstrual cycle. Mechanistically, ovulatory injury activates the extrinsic coagulation cascade, in which the activated thrombin cleaves the pro-HGFA to the activated-HGFA, which subsequently cleaves the pro-HGF to HGF. The FF has a high reserve of these coagulation cascade proteins and HGFA/HGF. The cascade is slowly and continuously activated by a tissue factor released in the FF and peritoneal fluid [36]. Thus, c-MET-expressing tissues, including the FTE and the endometriosis lesions, all receive the signal and may exhibit growth and invasion phenotypes [44]. Reportedly, c-MET, like IGF-1R, is highly expressed in the eutopic and the ectopic endometrial tissues [41,45]. Their signals promote the proliferation and invasion of the endometrial epithelial cells and stromal cells [39,41,46], playing important roles in the development of endometriosis and associated pelvic pain [39,47] (Figure 2).

### 3.6. High Concentrations of Soluble Extracellular Matrix (ECM) Proteins in the FF May Also Contribute to the Development of Endometriosis

In addition to the IGF-axis proteins and the HGF, extremely high concentrations of the ECM proteins are present in ovulatory FF [48]. They include fibronectin, vitronectin, laminin, serpin family D member 1 (SERPIND1), and periostin. The highly abundant ECM protein in FF leads to a steep concentration gradient between the FF, peritoneal fluid, and serum [49]. The follicle–blood barrier (FBB) between the granulosa cells lining the ovarian follicle and the surrounding theca cells traps molecules larger than 300 kDa in the ovarian follicle [50,51]. As shown in Table 1, the FF contains extraordinarily high concentrations of macromolecule-bound growth factors, sex hormones, and soluble ECM proteins [48]. Estrogen is the most concentrated molecule in the FF (990-fold and 2900-fold higher in the FF than in the peritoneal fluid and serum) and is essential for the initiation and development of endometriosis [52]. In addition, the concentrations of PAPP-A and HGF are 36-fold and 80-fold higher in the FF than in the PF, respectively; PAPP-A has a 500-fold higher concentration in the FF than in serum. Previously, we found that PAPP-A, which cleaves IGFBP 2/6, could release IGF2 to bind with its receptor IGF-1R on the FTE, and its concentration was positively associated with the cell transformation activities of the FF [21]. 

Soluble fibronectin and laminin are the 4th and 69th most abundant proteins in the FF [48]. They promote cell adhesion, seeding, migration, and invasion in different cancers, including ovarian cancer, through the integrins αvβ1 and αvβ3 [53,54]. Their levels in the FF are approximately 100-fold higher than that in serum. Other soluble ECM proteins, such as vitronectin (the 11th) [55], SERPIND1 (the 38th) [56], and periostin (the 112th) [57], also exhibit different oncogenic activities in ovarian cancer. Vitronectin promotes adhesion and chemoattraction in ovarian cancer cells via the α5β1 and αvβ3 integrins and the urokinase plasminogen activator surface receptor [55]. The SERPIND1 protein, which belongs to a family of serine protease inhibitors and is highly expressed in ovarian cancer tissues, confers poor prognoses in ovarian cancer [56]. It promotes ovarian cancer cell proliferation, epithelial-to-mesenchymal transition, migration, and invasion, and inhibits apoptosis via the PI3K/AKT pathway [56]. Whereas, periostin, overexpressed in the advanced and recurrent ovarian cancers, promotes adhesion, migration, and invasion in ovarian cancer and the human umbilical vein endothelial cells. It also promotes angiogenesis and xenograft metastasis when overexpressed in the HGSC cells [58]. Although the levels of vitronectin and periostin in the peritoneal fluid and the role of these ECM proteins in the endometriosis development are unknown, the common receptor, αvβ3 integrin, is abundantly expressed in the eutopic endometrium of an endometriosis patient as well as in normal endometrium [59,60].

### 3.7. Ovulation May Also Promote the Malignant Transformation of Ovarian Endometriosis

Although extra-ovarian endometriosis lesions are always benign, about 0.5% to 1% of ovarian endometriosis is associated with malignant neoplasia, most commonly, endometrioid carcinoma and clear-cell carcinoma [72]. About 36% of the ovarian clear-cell carcinomas and 19% of the ovarian endometrioid carcinomas are associated with endometriosis [73]. These endometriosis-associated ovarian cancers are believed to develop from ovarian endometriosis since a transition from the atypical endometriosis to the borderline malignancy or invasive carcinoma is frequently found. Moreover, driver mutations in *PIK3CA*, *KRAS,* and *ARID1A* have been found in the epithelium of ovarian and extra-ovarian endometriosis tissues and ovarian cancers associated with endometriosis [74,75]. Given the repeated exposure to ovulatory FF, the same mechanism of ovulation-induced carcinogenesis may also act on ovarian endometriosis. This explains why only the ovarian but not the extra-ovarian endometriosis is at risk of malignant transformation.

## 4. Missing Link 2: Retrograde Menstruation Is the Source of the Tissue-of-Origin in Endometriosis and Also Augments the Disease

The human menstrual cycle is an orchestrated series of physiological, cellular, and biochemical changes that are complexly regulated by hormones and involve the routine thickening of the endometrium in the preparation for embryo implantation. Menstruation is the cyclic breakdown of the endometrium in the absence of an embryo implantation, and it is one of the three (other than ovulation and parturition) physiological tissue injuries caused by acute inflammation [76]. Unexpectedly, 90% of all women who underwent laparoscopy during menstruation had a bloody peritoneal fluid, suggesting that retrograde menstruation is a common phenomenon in women [9]. This provides the basis of Sampson’s theory of the genesis of endometriosis, which states that the eutopic endometrium is seeded into the peritoneal cavity via retrograde menstruation [8].

Since vigorous regeneration of the endometrial tissue is required at each menstruation, the shed endometrial tissues in the menstrual flow are very likely to harbor stem/progenitor cells. It is up to these stem cells to establish the ectopic growth and development of the endometriosis. Meanwhile, the regenerative capacity of the endometrium is attributed to the stem or progenitor cells residing in the basalis layer which are not shed by the menstruation and provide a source to regenerate the functionalis layer each month [77]. In the eutopic endometrium of endometriosis women, there is a propensity for the localization of stem cell marker-positive basalis-like cells in the functional layer [78]. Additionally, the menstrual blood of endometriosis women also tends to harbor fragments of shed basalis tissue.

The mechanisms involving the survival and growth of the endometrial stem cells in the peritoneal environment are crucial for the development of endometriosis. These mechanisms involve (1) overcoming the apoptosis due to a loss of attachment (or anoikis resistance) after intraperitoneal shedding, (2) the attachment and proliferation on the peritoneum, (3) angiogenesis, and (4) stromal invasion. These mechanisms are identical to the intraperitoneal metastasis of ovarian HGSC. Moreover, patients with endometriosis have an increased risk of experiencing upper genital tract and peritoneal infections [79]. Mechanistically, the microbiome and oxidative stress introduced by the retrograde menstruation may cause microtrauma to the mesothelium lining, exposing the intercellular matrix and inducing acute inflammation [80]. These changes may contribute to the successful implantation of the ectopic endometrial tissues.

### Extrauterine Menstruation in Endometriosis Triggers Inflammatory Response

In addition to being the source of ectopic endometrial tissue, menstrual debris in the peritoneal cavity readily releases damage-associated molecular patterns (DAMPs) that trigger an innate immunity in women with or without endometriosis [81]. The DAMPs that are released by the activated mast cells and macrophages lead to the secretion of pro-inflammatory and pro-nociceptive mediators, exacerbating the inflammation and generating pain signals in endometriosis; thus, the symptoms of peritoneal inflammation persist and strengthen [82].

In retrograde menstruation, erythrocytes and their debris are engulfed by macrophages, thus inducing oxidative stress [83] and the release of pro-oxidative and pro-inflammatory factors, such as heme and iron [84]. Indeed, high levels of iron, ferritin, and hemoglobin have been found in the peritoneal fluid of women with endometriosis [84]. The release of iron and reactive oxygen species (ROS) has been implicated in the formation of deleterious free radicles where ferrous ion catalyzes the Fenton reaction, causing oxidative injury to cells [84]. Thus, ROS may also help with the destruction of the peritoneal mesothelium, allowing the attachment and growth of ectopic endometrial cells in the peritoneal cavity [85].

Similar to the mechanism of regeneration after ovulation, HGF is strongly expressed in the human endometrial epithelium during menstruation, stimulating the proliferation, migration, and lumen formation in human endometrial epithelial cells [86,87]. Studies have reported that HGF levels in the PF are elevated in women with endometriosis [88]. In addition, the IGF axis also plays an essential role in endometrial biology, with the IGFs being secreted by the stroma and their receptors being expressed by the endometrial epithelium [89]. These components of the regenerating machinery may be released into the peritoneal cavity from retrograde menses and help in the initiation and progression of endometriosis.

## 5. Missing Link 3: Hypermutation and Immune Escape in Endometriosis

### 5.1. Hypermutation Phenotype of Endometriosis

After menstrual shedding, the body vigorously regenerates the endometrium by activating and recruiting local [90] and bone marrow-derived [91] stem cells to the uterine lining and replenishes it by asymmetric cell division and clonal expansion. However, DNA replication during cell division is not without errors. Each round of replication creates thousands of DNA mismatches, leaving a mutation rate of about 10 bases per replication per genome [92]. By the nature of the asymmetric division, where one daughter cell is designated to retain the stemness, the stem cells in the endometrium would pass newly acquired mutations to their offspring indefinitely, including the derived endometriosis. Recently, whole-genome sequencing of 292 endometrial glands that were freshly microdissected from 28 women detected a high somatic mutation load of 1521 base substitutions or indels (ranging from 209–2833) per person [93].

This number of mutations was associated with the age of a woman, with a linear accumulation of 29 base substitutions per gland per year. Moreover, a higher mutation load was observed in both the eutopic and ectopic endometrial cells obtained from women with endometriosis (~25 mutations per megabase) [94], than that in women without endometriosis (8.7 mutations per megabase) [93]. This high mutation load surpassed that of common cancers [95,96,97] and was similar to that of the ultra-mutated group of endometrial cancer [98]. Interestingly, although these patients had a similarly high somatic mutation load of eutopic and ectopic endometrial cells, the latter carried a higher mutated allele frequency in the cancer-driving genes [74,99], indicating a cancer-like mechanism of clonal evolution and selection. Meanwhile, a whole-exome sequencing study of deep infiltrating endometriosis lesions (29 lesions from 24 patients) that were macrodissected (by tissue coring) from archived formalin-fixed paraffin blocs, reported a low mutation burden of 3.3 mutations per person [74].

### 5.2. Immune Escape in Endometriosis and the Possibility of Immune Checkpoint Blockade Therapy

The expression of neoantigens from the hypermutated genome renders the mutant cells to the cytotoxicity of immune surveillance. One crucial mechanism of this surveillance is to discriminate the normal, orthotopic, and hardly mutated “self” cells from the hypermutated and ectopically-localizing “non-self” cells, and eradicate the latter by the activation of cytotoxic T cells. This self-identification works largely through the T cell receptor (TCR)/antigen/MHC self-recognition complex and the PD-1/PD-L1 immune checkpoint. Once a cell is recognized as a “self” by the TCR/MHC complex, the PD-L1 ligand of the “self” cell binds to the PD-1 receptor on the T cell, and suppresses the T cell-mediated cytotoxicity, leading to an immune tolerance status. In contrast, neoantigen-carrying cancer cells or retrograded endometrial cells are not recognized as a “self”, thus bypassing the immune checkpoint and undergoing T cell-mediated cytotoxicity [100]. Given the ultra-mutation phenotype, retrograded endometrium from the endometriosis patients would rely on the activation of the PD-1/PD-L1 immune checkpoint to evade the immune surveillance during the development of endometriosis. Concurring with a previous report [101], Figure 3 demonstrated an example of the overexpression of PD-1 in the stroma (very likely in the immune cells) and PD-L1 in the epithelial cells of both the eutopic endometrium and ectopic endometriosis in an endometriosis patient. The cancer-like immune tolerance in endometriosis may predict the efficacy of a holistic treatment strategy by conducting an immune checkpoint blockade that has revolutionized cancer treatment [102]. Testing of the DNA mutation burden in the eutopic endometrium would predict the response, just as the documented practices of identifying hyper- or ultra-mutated endometrial cancers for treatment with immune checkpoint inhibitors do [103].

## 6. Implications in Pathogenesis and Clinical Management

One curiosity would be whether the ovulation engine drives the initial implant or the progression of endometriosis. Currently, there is no direct evidence to answer. However, given the stemness activation and clonal expansion activity of IGF2 and the clonogenicity activity of HGF in FF, we predict that ovulation would facilitate both the initiation and progression of endometriosis and we are currently conducting animal studies to prove this.

The double engines and single checkpoint theory, if proven to be true, may revolutionize the current strategies in the prevention and management of endometriosis. Firstly, the concept that pregnancy reduces the risk and attenuates the severity of endometriosis [104,105] can be enforced, since both the incessant ovulation and menstruation are overcome during pregnancy. The theory also firmly supports the conventional medication of oral contraceptives, which inhibits ovulation and reduces the amount of menstruation. The efficacy of oral pills is comparable to the much more expensive and less tolerable GnRH agonist and antagonists (Elagolix) [106]. The progestin preparations, such as dienogest (Vissane) and desogestrel (Cerazette), which cease both ovulation and menstruation after long-term use (https://endometriosisnews.com/cerazette-desogestrel/ (accessed on 8 June 2022) [107], are currently the most effective and cost-effective therapeutics for controlling and preventing the recurrence of endometriosis [108,109]. The inherent anti-inflammatory and anti-estrogenic activities of progesterone further enforce its effectiveness in addition to ovulation and menstruation inhibition [110]. Meanwhile, the well-established local progestin preparation Mirena, although working efficiently in inhibiting menstruation, does not alter the pituitary function and ovulation due to its low systemic distribution [111]. Clinical studies have shown that although Mirena worked equally effective as GnRHa in relieving endometriosis-associated pain [112], it was not as effective in preventing the recurrence of ovarian endometrioma [25].

The “checkpoint” in the proposed theory, i.e., the establishment of immune tolerance via the immune checkpoint, may open the way for an immune checkpoint blockade by targeting PD-1/PD-L1 or other molecules in the treatment of endometriosis. While half a dozen of the immune checkpoint inhibitors have been licensed for the treatment of a wide range of cancers that are characterized by genomic hypermutation and a high replication error rate [110], endometriosis, with identical phenotypes, may very likely respond similarly to the same treatment. Identical biomarkers, such as the PD-L1 expression in tissue sections, and a mismatch repair deficiency (identified by microsatellite instability) or high mutation load (identified by next-generation sequencing) in tissue DNA, can be used to predict the response and selection of eligible cases [113]. Moreover, a long-lasting response can be expected because of the presence of the active immunological memory [114].

## 7. Conclusions

The enigma of the pathogenesis of endometriosis can largely be explained by the proposed double engines and single checkpoint theory that consists of three pillars: (1) Ovulatory FF contains growth and cell remodeling factors that promote the genesis and development of endometriosis. (2) The retrograde menstrual flow seeds ectopic endometrial tissue to the peritoneal cavity and triggers the inflammation that promotes the severity of the disease. (3) The turning on of the immune checkpoint and evading the immune surveillance, allows for the survival of hypermutated and ectopically located endometriosis cells. These pillars, as summarized in Figure 4, not only explain the pathogenesis of endometriosis but may also open new opportunities and new strategies for the prevention and management of endometriosis.

## Figures and Tables

**Figure 1 biomedicines-10-01403-f001:**
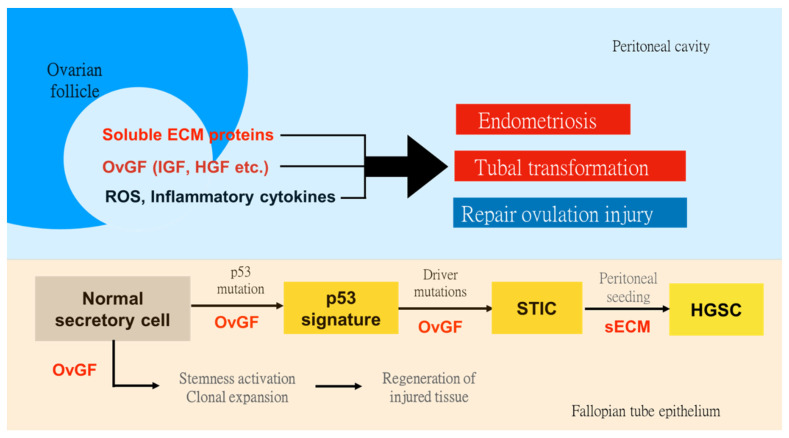
Growth and metastasis-promoting activities of the ovulatory follicular fluid in the peritoneal cavity that support tissue regeneration, malignant transformation, and the development of endometriosis. High levels of ROS, inflammatory cytokines, and growth factors (OvGFs), such as the IGF2 and HGF, in the ovulatory follicular fluid (FF), have been proved to be responsible for the malignant transformation of the fallopian tube fimbrial epithelium and regeneration of injured tissue caused by ovulation. ROS and inflammatory cytokines induce mutagenesis to create driver mutations; OvGFs promote clonal expansion by stemness activation and mitogenesis, and metastasis by enhancing attachment growth, migration, and invasion. Under these FF activities, the *TP53* mutation-initiated epithelial lesion (p53 signature) progresses to serous tubal intraepithelial carcinoma (STIC) and metastasizes to the ovary and peritoneum to grow high-grade serous carcinoma (HGSC). The same FF activities also mediate the regeneration of post-ovulatory injuries on the ovary and fimbria. It may also be responsible for the genesis and development of endometriosis by promoting mutagenesis, stemness, survival, attachment, growth, and peritoneal spreading and invasion of retrograded endometrial cells, as well as the preexisted endometriosis cells which are samely exposed by FF.

**Figure 2 biomedicines-10-01403-f002:**
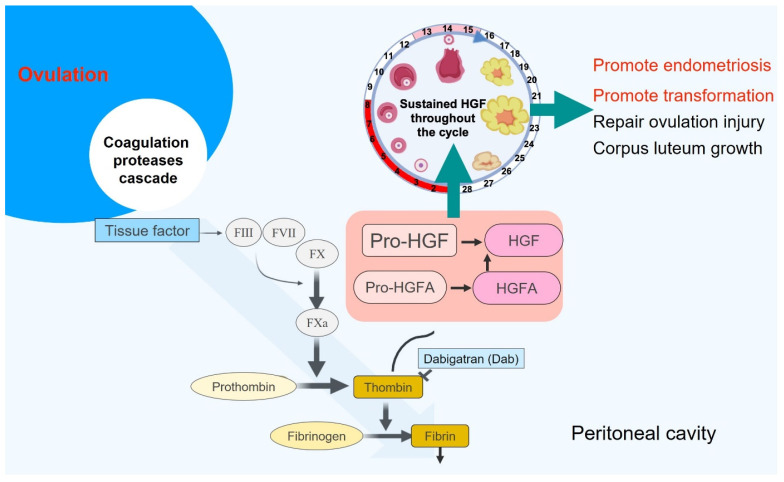
The coagulation cascade/HGF machinery in the follicular fluid exhibits prolonged growth-related activity that may promote the genesis of endometriosis. Follicular fluid (FF) exhibits prolonged transformation and regeneration activities before and after mixing with the peritoneal fluid. The coagulation cascade-activated hepatocyte growth factor (HGF) is responsible for this sustained activity. FF harbors high levels of coagulation cascade proteases and proform (pro-) HGFA and pro-HGF, such that the active form of HGF can be constitutively supplied by the cascade cleavage of pro-HGFA and pro-HGF. This prolonged HGF activity, which lasts for the whole menstrual cycle, promotes the proliferative repair of the ovarian surface and fallopian tube fimbrial epithelia, as well as the growth of the corpus luteum. The same activity also inadvertently promotes the malignant transformation of the fimbrial epithelium [36] and, likely, the development of endometriosis.

**Figure 3 biomedicines-10-01403-f003:**
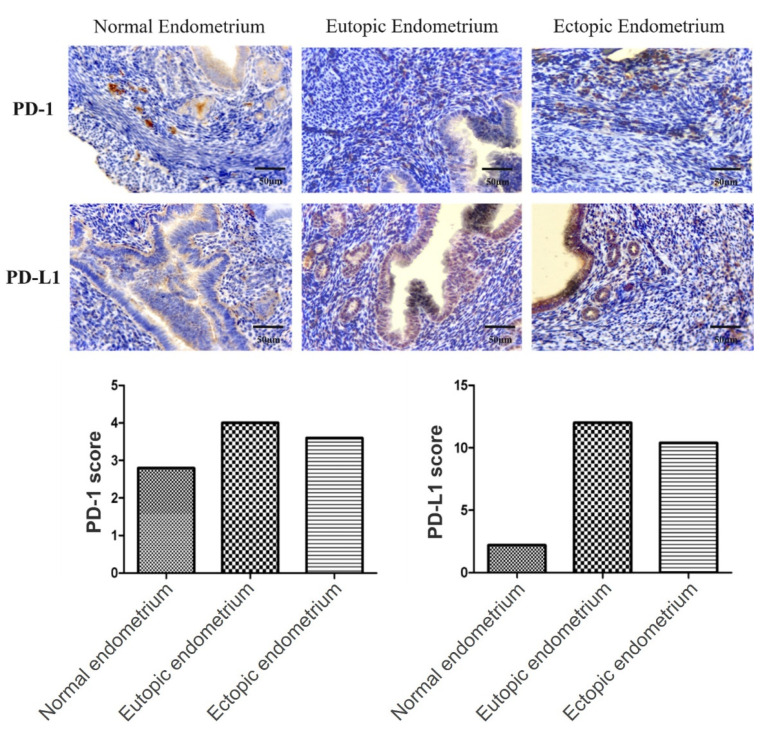
An example of overexpression of PD-L1 in endometriosis. Immunohistochemistry of PD-1 and PD-L1 in the eutopic endometrium and ectopic endometrium (endometriotic lesions) of an endometriosis patient and in the endometrium of a healthy woman is shown. The immunoreactive scoring, in which a product of the IHC staining intensity (0–3) and the percentage of positively stained cells (0, 0%; 1, 1–10%; 2, 11–50%; 3, 51–80%, and 4, ≥80%) was calculated, are shown in the lower panel.

**Figure 4 biomedicines-10-01403-f004:**
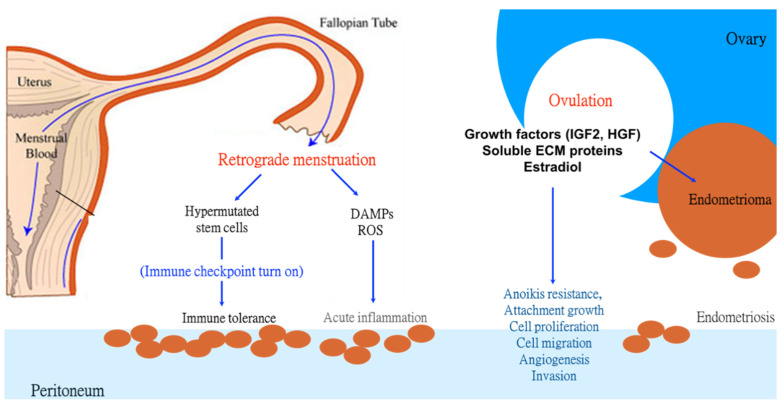
The double engines and single checkpoint theory of the genesis of endometriosis. Retrograde menstruation sources stem cells from the broken-down endometrial tissue. It also supplies immunogenic and tissue-damaging substances, such as damage-associated molecular patterns (DAMPs) and reactive oxygen species (ROS), to trigger acute inflammation and possibly mutagenesis in endometriosis. Ovulation is the other engine that drives the development of endometriosis. Fueled by mutagenic and carcinogenic components that drive malignant transformation, such as growth factors, soluble extracellular matrix (sECM) proteins, and estradiol in the ovulatory follicle, ovulation can induce or augment transforming activities that also act on the retrograded endometrial cells or preexisting endometriosis cells, promoting the development of endometriosis: anoikis resistance, attachment growth, cell proliferation, migration, and invasion, as well as angiogenesis. Meanwhile, the endometriotic cells inherit the ultra-mutated genome from the stem cells in the eutopic endometrium. By up-regulating the PD-1/PD-L1 immune checkpoint, they acquire an immune tolerant state for long-term survival and growth in ectopic environments.

**Table 1 biomedicines-10-01403-t001:** Concentration gradients of representative hormones, growth factors, and extracellular matrix proteins in the FF, PF, and serum.

	E2, pg/mL	PAPP-A, mIU/L	HGF, ng/mL	Fibronectin, ug/mL	Laminin, ng/mL
Serum	~170 (80–350)	1.7 +/− 0.3	0.86 +/− 0.04	0.2~0.4	0.78
	[49]	[61]	[62]	[63]	[63]
FF	496 K +/− 144 K	810 +/− 290	74 +/− 2.9	~70	500
	[64]	[61]	[62]	[63]	[65]
PF	~500	31.4 +/− 15.4	0.92 +/− 0.11	60 +/− 4.92	960 (24–1502)
After ovulation	44,000	Nil	Nil	Nil	Nil
Secretory phase	~900 (early)	44.5 +/− 15.7 9	Nil	Nil	Nil
Follicular phase	~120	22.7 +/− 12.1	Nil	Nil	Nil
	[38,49]	[66]	[67,68]	[69,70]	[71]
FF/Serum	2900×	500×	86×	130×	86×
FF/PF	990×	36×	80×	1.2×	0.52×

FF, follicular fluid; PF, peritoneal fluid; [], reference; Nil: No data.

## Data Availability

Not applicable.

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
