# Peer review of "The Double Engines and Single Checkpoint Theory of Endometriosis"

_biomedicines, 2022, doi:10.3390/biomedicines10061403_

Round 1

Reviewer 1 Report

This is the review of the pathogenesis of endometriosis, focusing on ovulation and retrograde menstruation, well known risk factor of endometriosis.  This review is well-written on an interesting and developing topic. 

There are several concerns to mislead readers, which should be improved before publication. 

  1. The meaning of “missing link” is ambiguous. In P2 L15, it is defined as “missing links between the development and persistence of endometriosis”. Readers may misunderstand that the authors hypothesized the mechanism of development of endometriosis is different from that of persistence, and you would discuss the evidence to link them.
  2. On P3 L26, authors claimed that “estrogen alone does not support the growth of endometriosis” since add-back therapy with GnRH agonist is effective. However, the reason why estrogen level of add-back therapy is far less than physiological level during ovulation. In addition, it is well-known that estrogen replacement therapy has a risk to develop endometriosis in the menopausal period. 
  3. The meaning of 1-4(P3 L48) is unclear. If authors would like to emphasis this part, the importance of cell stemness and clonal expansion in the pathogenesis of endometriosis should be discussed. It is true endometriosis behaves like a cancer, but endometriosis is not tumor but normal tissue on ectopic lesion. Thus, it seems strange to use most of part are used for the discussion on ovarian carcinogenesis. 
  4. “pregnancy is the best treatment for endometriosis” (P9 L16) is incorrect. It is true null pregnancy is a risk factors of endometriosis, however in a part of ovarian endometrioma patient their size increase during pregnancy. Authors should show clinical evidence that pregnancy reduce the risk of endometriosis (for example, pain score comparison before and after pregnancy or recurrence rate of endometrioma after pregnancy). 
  5. The names of medicine as below is incorrect. Dienoest (P3 L30) -> dienogest. Marveln(P9 L21) is a medical compound of desogestrel and ethinylestradiol, thus it is not proper for the example of progestin preparations. 

Author Response

(1)The meaning of “missing link” is ambiguous. In P2 L15, it is defined as “missing links between the development and persistence of endometriosis”. Readers may misunderstand that the authors hypothesized the mechanism of development of endometriosis is different from that of persistence, and you would discuss the evidence to link them.

Thanks to the comment of reviewer the sentence has been revised from “Moreover, recent studies have implied missing links between the development and persistence of endometriosis.” to “Given these risk factors, recent studies have implied missing links between the retrograde menstruation and the genesis and progression of endometriosis.’

(2)  On P3 L26, authors claimed that “estrogen alone does not support the growth of endometriosis” since add-back therapy with GnRH agonist is effective. However, the reason why estrogen level of add-back therapy is far less than physiological level during ovulation. In addition, it is well-known that estrogen replacement therapy has a risk to develop endometriosis in the menopausal period.

Yes, I got your point, but I would like to believe that it is a matter of estrogen dose and level. Estrogen add back resume the E2 to a non-ovulatory physiological level, So does the postmenopausal estrogen replacement therapy. In the presence of ovulation and menstruation, this level of E2 would promote endometriosis growth. In the add-back scenario, since ovulation and menstruation are suppressed by GnRHa, therefire there is no increase of disease recurrence.

In the postmenopausal HRT, the concern is that exogenous estrogen could reactivate the pre-existing endometriotic foci. However, there is no solid evidence of this concern. Kadri (https://pubmed.ncbi.nlm.nih.gov/19160262/) and Zanello (https://pubmed.ncbi.nlm.nih.gov/31416164/) have systemically reviewed literatures and found only two randomized controlled trials and one retrospective observational study on the hormone therapy for women with endometriosis in post-surgical (BSO) menopause. Neither of these studies found statistically difference between the treatment and control group, nor was there a confirmation of recurrence or exacerbation of endometriosis.

(3)The meaning of 1-4(P3 L48) is unclear. If authors would like to emphasis this part, the importance of cell stemness and clonal expansion in the pathogenesis of endometriosis should be discussed. It is true endometriosis behaves like a cancer, but endometriosis is not tumor but normal tissue on ectopic lesion. Thus, it seems strange to use most of part are used for the discussion on ovarian carcinogenesis.

 Endometriosis in the pelvis does not grow a “tumor” but ovarian endometriosis typically grow the endometrioma. Endoemtrioma is the benign tumor and may subjected to malignant transformation, progressing from borderline endometroid tumor to endometrioid carcinoma or clear cell carcinoma of the ovary. Transitional histology of endometriosis, borderline malignancy, invasive carcinoma is typically found, suggesting the progressive evolution of malignant transformation (https://pubmed.ncbi.nlm.nih.gov/29884083/; https://pubmed.ncbi.nlm.nih.gov/29241974/).  I assume, the reason why only the ovarian endometriosis but not the endometriosis elsewhere can grow a tumor and have the potential to undergo malignant transformation is because it’s vicinity to the ovulation site.

 (4)“pregnancy is the best treatment for endometriosis” (P9 L16) is incorrect. It is true null pregnancy is a risk factors of endometriosis, however in a part of ovarian endometrioma patient their size increase during pregnancy. Authors should show clinical evidence that pregnancy reduce the risk of endometriosis (for example, pain score comparison before and after pregnancy or recurrence rate of endometrioma after pregnancy).

 Agree with the point. We changed the phrase“ Firstly, the concept that pregnancy is the best treatment for endometriosis can be enforced.“  to “ Firstly, the concept that pregnancy reduces the risk and attenuates the severity of endometriosis can be enforced,“

 (5)The names of medicine as below is incorrect. Dienoest (P3 L30) -> dienogest. Marveln(P9 L21) is a medical compound of desogestrel and ethinylestradiol, thus it is not proper for the example of progestin preparations.

 Thanks for the careful corrections. We corrected the phrase as: “Progestin preparations, such as dienogest (Vissane) and desogestrel (Cerazette) that cease both ovulation and menstruation after long-term use,…”

https://endometriosisnews.com/cerazette-desogestrel/; https://pubmed.ncbi.nlm.nih.gov/28266234/

Reviewer 2 Report

The authors point out that endometriosis remains an unclear entity and discuss the theory of pathogenesis of endometriosis. They present current pieces of evidence and therapeutic applications of the theory.

The double engines are ovulation (ovulatory FF) and menstruation (retrograde mehstrual flow) that drive the development of endometriosis. And development of endometriosis, because of neoantigen overload, activates the PD-1/PDL1 “self-tolerance” immune checkpoint to evade the immune surveillance.

This article is interesting and informative for gynecologists, pathologists and researchers. The most important point of this article is how to clarify the pathogenesis of endometriosis. For that purpose, they explained in an easy-to-understand manner using figures.

It might be improved on following issues.

1. P.9, Figure 3. (1) In the photos, are the positively stained cells glandular epithelial cells and/or stromal cells of endometriosis?

(2) The bar chart: The bars to the right of both the bar graphs (PD-1 and PD-L1) are not eutopic endometrium but ectopic endometrium.

(3) The immunoreactive scoring: Total PD-1 score and PD-L1 score should be 0-7 (staining intensity: 0-3, percentage of positively stained cells: 0-4). However, in this figure 3. Why PD-L1 score of the vertical axis unit 0-15?

2. Dose the author’s hypothese apply to endometriosis of other organs (such as lung, gastrointestinal and skin, etc.)?

Author Response

It might be improved on following issues.

1.P.9, Figure 3.

(1) In the photos, are the positively stained cells glandular epithelial cells and/or stromal cells of endometriosis?

Thanks for the questioning and the concern of the differential ligand/receptor expression. In the IHC, PD-L1 was expressed in the epithelium and PD-1 was expressed in the stroma. We revised the description accordingly: “ Figure 3 demonstrated an example of overexpression of PD-1 in the stroma and PD-L1 in the epithelium of both the eutopic endometrium and ectopic endometriosis lesion in an endometriosis patient.“

(2) The bar chart: The bars to the right of both the bar graphs (PD-1 and PD-L1) are not eutopic endometrium but ectopic endometrium.

We corrected the labelling error. Thanks a lot.

 (3) The immunoreactive scoring: Total PD-1 score and PD-L1 score should be 0-7 (staining intensity: 0-3, percentage of positively stained cells: 0-4). However, in this figure 3. Why PD-L1 score of the vertical axis unit 0-15?

We adopted the immunoreactive score (IRS) system which gives a range of 0–12 as a product of multiplication between positive cells proportion score (0–4) and staining intensity score (0–3)[https://www.ncbi.nlm.nih.gov/pmc/articles/PMC4260254/].

2. Dose the author’s hypothese apply to endometriosis of other organs (such as lung, gastrointestinal and skin, etc.)?

Yes it may! We are exploring more evidences showing the ovary is not only an endocrine organ to secret hormones, it also secret growth factors to the systemic circulation.

Reviewer 3 Report

It an excellent manuscript regarding new insight into pathogenesis of endometriosis. I read it with pleasure. The article should be accepted for publication.

I would add 1 short paragraph regarding ovarian cancer risk in patients with endometriosis. It is commonly known that endometriosis-associated ovarian cancer, most commonly clear cell carcinoma, is believed to develop from ovarian endometrial cysts. Driver mutations in i.e. PIK3CA, KRAS, ARID1A, have been found in the epithelium of ovarian and extraovarian endometriosis tissue and ovarian cancers associated with endometriosis (i.e., clear cell and endometrioid type). Please present shortly this problem. You mentioned possible anti-PD-1 and PD-L1 treatment in endometriosis and this kind of immunotherapy is extensively studied in ovarian cancer. 

Please check the manuscript for some minor spelling mistakes, i.e. page 9, line 20 (inexpensive -expensive)

Author Response

REVIEWER 3:

(1) I would add 1 short paragraph regarding ovarian cancer risk in patients with endometriosis. It is commonly known that endometriosis-associated ovarian cancer, most commonly clear cell carcinoma, is believed to develop from ovarian endometrial cysts. Driver mutations in i.e. PIK3CA, KRAS, ARID1A, have been found in the epithelium of ovarian and extraovarian endometriosis tissue and ovarian cancers associated with endometriosis (i.e., clear cell and endometrioid type). Please present shortly this problem. You mentioned possible anti-PD-1 and PD-L1 treatment in endometriosis and this kind of immunotherapy is extensively studied in ovarian cancer.

Echoing the reviewer’s suggestion, we added a new paragraph at the end of Missing link 1:

Ovulation may also promote the malignant transformation of ovarian endometriosis

Although extra ovarian endometriosis lesions are always benign, about 0.5% to 1% of ovarian endometriosis are associated with malignant neoplasia, most commonly endometrioid carcinoma and clear cell carcinoma [https://pubmed.ncbi.nlm.nih.gov/29241974/]. About 36% of ovarian clear-cell carcinoma and 19% of ovarian endometrioid carcinoma are associated with endometriosis [https://pubmed.ncbi.nlm.nih.gov/24518590/]. These endometriosis-associated ovarian cancers are believed to develop from ovarian endometriosis since a transition from the atypical endometriosis to the borderline malignancy or invasive carcinoma is frequently found. Moreover, driver mutations in the PIK3CA, KRAS, ARID1A, have been found in the epithelium of ovarian and extra ovarian endometriosis tissue and ovarian cancers associated with endometriosis [https://pubmed.ncbi.nlm.nih.gov/28489996/; https://www.ncbi.nlm.nih.gov/pubmed/30110635]. Given the repeated exposure to ovulatory FF, the same mechanism of ovulation-induced carcinogenesis may also act on ovarian endometriosis. This explains why only the ovarian but not the extra ovarian endometriosis is at risk of malignant.

(2) Please check the manuscript for some minor spelling mistakes, i.e. page 9, line 20 (inexpensive -expensive)

Thanks for the careful corrections.

Round 2

Reviewer 1 Report

Although the authors responded well to our comments, several minor change are still required.  

(2) Authors cite the meta-analysis of HRT, but they use both estrogen and progestins.  Conbination of estrogen and progestin such as OC reduce the risk of endometriosis, since progestin has therapeutic effect on endometriosis.  Before the ovulation, the level of E2 is about 200pg/ml, which is far larger than the level during estrogen add-back therapy (about 50pg/ml) .  Thus the authors could not claim "estrogen alone does not support the growth of endometriosis".  

(4)  Authors should cite the clinical evidence that pregnancy reduces the risk of endometriosis.   

Author Response

REVIEWER 1 (Round 2):

Although the authors responded well to our comments, several minor change are still required.  

(2) Authors cite the meta-analysis of HRT, but they use both estrogen and progestins.  Conbination of estrogen and progestin such as OC reduce the risk of endometriosis, since progestin has therapeutic effect on endometriosis.  Before the ovulation, the level of E2 is about 200pg/ml, which is far larger than the level during estrogen add-back therapy (about 50pg/ml) .  Thus the authors could not claim "estrogen alone does not support the growth of endometriosis".  

Response:

Agree with the suggestion. We removed the touch statement of "estrogen alone does not support the growth of endometriosis", and changed the phrase as: “Thus, ovulation and menstruation factors seem to be more important. “

Meanwhile, we went to the literature for serum E2 level in women at different menstrual cycle days and in women taking oral conjugated estrogen (Premarin 0.625 mg daily) (references listed in below). As shown in the WIKIMEDIA figure which summarized three studies (https://commons.wikimedia.org/wiki/File:Estradiol_levels_across_the_normal_menstrual_cycle_in_women.png) and a representative reference in 2021 Scientific Report [https://www.ncbi.nlm.nih.gov/pmc/articles/PMC7878477/], the median concentration of E2 before ovulation period is around 50 to 100 pg/mL. This level is the same as the level in women taking 0.625 mg conjugated estrogen, the typical dose of estrogen add-back.

˙Abbott (2009). ARCHITECT Estradiol Assay (PDF).

Ë™(2006). "Establishment of detailed reference values for luteinizing hormone, follicle stimulating hormone, estradiol, and progesterone during different phases of the menstrual cycle on the Abbott ARCHITECT analyzer". Clin. Chem. Lab. Med. 44 (7): 883–7. DOI:10.1515/CCLM.2006.160. PMID 16776638.

Ë™(February 2008). "Mass spectrometric and physiological validation of a sensitive, automated, direct immunoassay for serum estradiol using the Architect". Clin. Chim. Acta 388 (1-2): 99–105. DOI:10.1016/j.cca.2007.10.020. PMID 18023274

Ë™(February 2021) “Serum estradiol level according to dose and formulation of oral estrogens in postmenopausal women. doi: 10.1038/s41598-021-81201-y. PMID: 33574350

(4)  Authors should cite the clinical evidence that pregnancy reduces the risk of endometriosis.   

Response:

Yes. We cited the following two references in the revised phrase “ Firstly, the concept that pregnancy reduces the risk and attenuates the severity of endometriosis can be enforced,“ (page 10, line 9). https://pubmed.ncbi.nlm.nih.gov/26373341/ Parity and endometrial cancer risk: a meta-analysis of epidemiological studies https://pubmed.ncbi.nlm.nih.gov/29471493/ The effect of pregnancy on endometriosis-facts or fiction?
